# Changes in Healthy Behaviors and Meeting 24-h Movement Guidelines in Spanish and Brazilian Preschoolers, Children and Adolescents during the COVID-19 Lockdown

**DOI:** 10.3390/children8020083

**Published:** 2021-01-26

**Authors:** José Francisco López-Gil, Mark S. Tremblay, Javier Brazo-Sayavera

**Affiliations:** 1Departamento de Actividad Física y Deporte, Facultad de Ciencias del Deporte, Universidad de Murcia (UM), 30720 San Javier, Spain; josefranciscolopezgil@gmail.com; 2Healthy Active Living and Obesity Research Group, CHEO Research Institute, Ottawa, ON K1H 8L1, Canada; mtremblay@cheo.on.ca; 3Department of Sports and Computer Science, Universidad Pablo de Olavide (UPO), ES-41013 Seville, Spain; 4PDU EFISAL, Centro Universitario Regional Noreste, Universidad de la República (UDELAR), 40000 Rivera, Uruguay

**Keywords:** physical activity, sedentary behavior, sleep, healthy lifestyle, coronavirus

## Abstract

Background: The aim of this study was to assess changes in physical activity, screen time and sleep duration of preschoolers, children and adolescents and the prevalence of meeting the 24-h movement guidelines during the lockdown caused by COVID-19 in a sample from Spain and Brazil. Methods: A total of 1099 preschoolers, children and adolescents (aged 3–17 years) from Spain (12.1 ± 4.6 years) and Brazil (10.7 ± 4.3 years) were included. An online survey was created and distributed in each country using a snowball sampling strategy. This online survey was completed by parents (mother/father/responsible guardian). Results: The proportion of the sample who met the PA and ST recommendations decreased during the COVID-19 lockdown in both Spanish and Brazilian samples (*p* < 0.001), while sleep duration increased (*p* < 0.001). The proportion of the sample meeting the overall 24-h movement guidelines was very low before the lockdown (Spain 3.0%; Brazil 11.7%) and even worse during the lockdown (Spain 0.3%; Brazil 7.5%). Conclusions: The prevalence of preschoolers, children and adolescents in both the Spain and Brazil samples meeting the 24-h movement guidelines during COVID-19 restrictions was as low as previous studies in other countries. Efforts to protect and support healthy behaviors of young people during a period of pandemic restrictions need to be a priority.

## 1. Introduction

The coronavirus disease 2019 (COVID-19) pandemic has resulted in profound health, social and economical consequences. Among them is the immediate interruption of child care and school programs for preschoolers, children and adolescents around the world who by order had to stay in their homes during the lockdown aimed at containing the spread of COVID-19 [1]. The pandemic significantly interrupted normal activities around the world; for example, “stay-at-home” orders and lockdown periods have increased the use of digital entertainment [2].

Moreover, 24-h movement guidelines recommend that preschoolers, aged 3–4 years, should accumulate at least 180 min of physical activity (PA), engage in no more than 1 h sedentary screen time (ST) and have 10–13 h good-quality sleep per day [3]. For both children and adolescents (5–17 years), it is recommended to participate in at least 60 min of moderate-to vigorous-intensity physical activity (MVPA), engage in no more than 2 h sedentary recreational screen time and have 8–11 h good-quality sleep each day [4]. Although the benefits of complying with PA, ST and sleep recommendations have been shown independently [5,6,7], accumulating evidence shows that the composition of movement behaviors across the whole day matters for optimal health [8,9]. Kuzik et al. [10] suggested that adequate combinations of movement behaviors (e.g., PA, sedentary behavior and sleep duration) could hold great importance for maintaining optimal health during the early years of development. Similarly, Saunders et al. [8] reported that children and youth meeting the 24-h movement guidelines (high PA/high sleep/low sedentary behavior) generally present more favorable indicators of adiposity and cardiometabolic health, when compared with those not meeting these recommendations. The current evidence indicates that the composition of movement behaviors within a 24-h period may have important implications for health at all ages and that meeting the current 24-h movement guidelines is associated with a number of desirable health indicators in children and youth [9]. However, COVID-19-related restrictions likely exacerbate the current public health problems of low levels of PA and high prevalence of sedentary behaviors in the young population [11].

In order to identify how best to support families in the COVID-19 context, it is necessary to understand how movement behaviors have changed and impacted families. While studies performed in Canada [12,13], China [14], Italy [15], Croatia [16] and Spain [17,18] have analyzed the changes in PA, ST and sleep duration in young people, to the best of our knowledge, this is the first study that provides information on changes in meeting the 24-h movement guidelines in Spain and Brazil during the COVID-19 lockdown. Hence, the aim of this study was to assess the change in PA, ST and sleep duration of preschoolers, children and adolescents and the prevalence of meeting the 24-h guidelines during the lockdown caused by COVID-19 in a sample from Spain and Brazil.

## 2. Materials and Methods

Parents/guardians of children aged 3–17 years were recruited via social networks. An online survey was created and distributed in each country using a snowball sampling strategy. The online survey required around 15 min to complete. Before completing the survey, information about the aims of the research was given and informed consent was requested. Data were collected for 15 days in both countries (in Spain from 29th March and in Brazil from 14th April, both in 2020). Of the initial 1263 respondents from Spain and Brazil, 143 participants were removed because they were <3 years old or >17 years old. An additional 21 participants were removed because of missing values. Thus, data from 1099 respondents were included in the final analysis. This research was approved by the Ethical Committee of the Universidade Tecnológica do Paraná (UTFPR) (CAAE: 32023220.8.0000.5547; approval number: 4.275.232) and the Universidad Católica de Murcia (UCAM) (code: CE112001).

Respondents were asked to complete the survey if they had been isolated during the previous week and provided general information in the first part of the questionnaire on the restriction situation during the previous week. Parents’ information about nationality, socioeconomic status (by the Family Affluence Scale—FAS-III [19]), educational level, age and sex of their children was required. Anthropometric data were reported by parents about their children. Height was reported in centimeters and weight in kilograms. Both the z-score for BMI and the classification of overweight/obesity were calculated according to the WHO criteria [20,21].

The measurement of PA was based on the following question: ‘‘Normally, how many days was your child physically active for a total of at least 60 min?’’. This measure has shown to have good reliability and validity [22]. Response options ranged from 0 to 7 days per week, in 1-day increments. Meeting the PA recommendation was defined as 60 min of MVPA per day 7 days per week. This question was also asked for the COVID-19 lockdown period; however, the reliability was not assessed for the question during this period.

ST was assessed by asking respondents to indicate the time that their child spent in different sedentary screen-based pursuits. The following questions were asked for weekdays, weekends and during the COVID-19 lockdown: (a) “How many hours a day, in your child’s free time, do they usually spend watching TV, videos (including YouTube or similar services), DVDs, and other entertainment on a screen?”; (b) “How many hours a day, in your child’s free time, do they usually spend playing games on a computer, games console, tablet, smartphone or other electronic device (not including moving or fitness games)?”; and (c) “How many hours a day, in your child’s free time, do they usually spend using electronic devices such as computers, tablets or smartphones for other purposes (e.g., homework, emailing, tweeting, Facebook, chatting, surfing the internet)?”. A weighted (5 weekdays and 2 weekend days) sum of the three questions was computed. ST was categorized as follows: preschoolers (“not meeting ST guidelines”: >1 h/d; “meeting the ST guidelines”: ≤1 h/d); and children/adolescents (“not meeting ST guidelines”: >2 h/d; “meeting the ST guidelines”: ≤2 h/d). This categorization was based on WHO guideline recommendations for ST for children under 5 years old [3] and on the Canadian guidelines on ST for children and youth [4].

Sleep duration was assessed by asking respondents for weekdays and weekend days separately: “What time does your child usually go to bed?” and “What time does your child usually get up?”. These questions were also asked for the COVID-19 lockdown period. The average daily sleep duration was computed for each participant as follows: [(average nocturnal sleep duration on weekdays × 5) + (average nocturnal sleep duration on weekends × 2)]/7. Responses within the range of 10–13 h for 3–4-year-olds, 9–11 h for 5–13-year-olds and 8–10 h for 14–17-year-olds were categorized as “meeting sleep guidelines,” and participants out of these ranges were classified as “not meeting sleep guidelines”, based on WHO international guidelines for early years [3] and 24-h movement behavior sleep recommendations [23].

Age, sex (males/females), breadwinner’s educational level (incomplete primary education, complete primary education, incomplete secondary education, complete secondary education, incomplete higher education or complete higher education), socioeconomic status (SES; low SES, medium SES or high SES) [19] and BMI (z-score) [20,21] were included into the model as covariates. The selection of these covariates was based on the correlates pointed out in the scientific literature [9].

Descriptive data were expressed as mean (standard deviation) for continuous variables and number (percentage) for categorical variables, except where specified otherwise. The proportion of preschoolers, children and adolescents meeting the 24-h guidelines and the component individual behaviors from each country was calculated. Differences between Spanish and Brazilian participants in sociodemographic and anthropometric information were analyzed using the chi-square (χ2) test for categorical variables and Student’s t test for continuous variables. A generalized linear mixed model was applied to assess the differences on changes in PA, screen time and sleep duration between each age group. Differences in meeting the 24-h movement guidelines and the individual component recommendations were examined by an independent McNemar’s test (categorical variables). All the analyses were carried out with the software Statistical Package for Social Sciences (SPSS, IBM Corp., Armonk, NY, USA), and a *p*-value of 0.05 denoted statistical significance.

## 3. Results

Table 1 shows the descriptive data of both Spanish and Brazilian study participants. The Spanish participants were older on average (M = 12.1 ± 4.6) than the Brazilian participants (M = 10.7 ± 4.3) (*p* < 0.001). A higher number of participants with high SES was observed in the Spanish sample (22.4% vs. 14.7%) (*p* < 0.001). Regarding educational level, a higher number of parents with complete higher education was reported in the Brazilian sample (70.9 vs. 31.0) (*p* < 0.001).

Figure 1 illustrates the mean (aggregated and stratified by age group) before and during the COVID-19 lockdown for the amounts of PA, ST and sleep duration. In general, fewer days of engaging in PA were shown in both Spanish and Brazilian participants (*p* < 0.001). ST during the period of imposed isolation was higher compared to the period before the lockdown for all age groups in both Spanish and Brazilian samples (*p* < 0.001). Sleep duration was higher during the COVID-19 lockdown in both samples (*p* < 0.001). Details regarding changes in these behaviors (stratified by age group) are shown in Appendix A.

Figure 2 illustrates the proportions meeting the 24-h movement guidelines and the various components in both Spanish and Brazilian populations. The proportion of participants who met the PA and ST guidelines decreased during the COVID-19 lockdown in both Spanish and Brazilian samples (*p* < 0.001), while sleep duration increased (*p* < 0.001). Additional results about the changes in meeting the 24-h movement guidelines according to the different age groups are available in Appendix A.

## 4. Discussion

The aim of the present study was to assess changes in PA, ST and sleep duration in preschoolers, children and adolescents and the prevalence of meeting the 24-h movement guidelines during the lockdown caused by COVID-19 in a sample from Spain and Brazil. Our findings show a decrease in the number of days per week with 60 min of MVPA during the COVID-19 lockdown in both Spanish and Brazilian samples in comparison with before the lockdown. In contrast, ST and sleep duration increased. A very low proportion of participants met the 24-h movement guidelines in both Spain (0.3%) and Brazil (7.5%). Our findings support those of others who reported a decrease in the time spent in PA [14,17,18] and in meeting PA recommendations [12,14].

Research findings demonstrate a dangerous downward trend in habitual PA levels among children during the COVID-19 pandemic [12,13,17]. Social restrictions including remote learning and “shelter-at-home” recommendations have made it difficult for children and adolescents to engage in physical education, sports or other forms of school-related or community-based organized PA [24]. A decline in outdoor play could contribute to diminished PA [13].

Our findings also show an increase in the ST of preschoolers, children and adolescents in both Spain and Brazil during the COVID-19 lockdown. These results match with other studies performed with information about the COVID-19 lockdown [12,13,14,16,17,18]. This observation could be explained by the circumstances of the COVID-19 lockdown, which could facilitate the “displacement theory”. This theory suggests that screen time may displace the time spent engaging in PA. This hypothesis has been supported longitudinally, with data that found that children who exceed the amount of ST (> 120 min per day) at age 6 were less active and had higher body mass indices at ages 8 and 10 years than children who at age 6 watched less television [25]. Parenting practices and children’s screen time during the COVID-19 pandemic could, at least partially, also justify these results, as a study on Turkish children has shown [26]. Another possible explanation for these results may be the lack of perceived capability of parents to restrict their children’s ST, as Guerrero et al. [12] showed in their study. Regardless of the reason, the dramatic rise in sedentary ST during the COVID-19 pandemic is concerning and strategies to mitigate this excessive behavior need to be established and implemented.

Our results show that sleep duration increased during the COVID-19 lockdown, especially in adolescents, also increasing the percentage of participants who met the sleep recommendations. This finding matches the study performed by Moore et al. [13] in Canada, who reported an increase in sleep duration of Canadian children and adolescents (based on a Likert scale reported by parents) and also reported by López-Bueno et al. [17], in a Spanish sample. Furthermore, Pietrobelli et al. [15] showed that sleep duration increased by more than half an hour daily during the COVID-19 lockdown among Italian obese children. Another study carried out on Chinese teenagers highlighted problems with sleeping, staying asleep or sleeping too much in about 40% of the participants [27]. In addition, it has been reported that sleep routines have been unstable, with no fixed timetables established during the COVID-19 pandemic for children and adolescents [24]. Due to more irregular timetables linked to the COVID-19 lockdown, preschoolers, children and adolescents may be sleeping more and, consequently, meeting sleep recommendations [24]. This slight increase could also be explained by the absence of travel to school, creating more discretionary time [24]. The higher increase in sleep time in adolescents could be due to the circadian shift towards eveningness that a lack of schedules may have caused on them [28]. However, these results need to be interpreted with care. First, it must be considered that human circadian rhythms have been shown to fluctuate with the seasons [29], and this fact may have influenced results, since during the COVID-19 lockdown in Spain, it was spring, while in Brazil, it was autumn. Second, although this reported increase in sleep duration seems to be a positive health-related finding, there is a need to control sleep habits in the younger population [30,31] (e.g., appropriate timing, regularity, sleep quality or sleep fragmentation).

Findings from this study suggest a significant decrease in the proportion of Spanish and Brazilian preschoolers, children and adolescents meeting the 24-h movement guidelines during the COVID-19 lockdown. This observation is in agreement with other studies [12,13]. However, caution is required to interpret these results. There are few studies analyzing meeting the 24-h movement guidelines during the COVID-19 lockdown and most of them relied on self-reported data by parents. In addition, the restrictions imposed by each country’s government during the COVID-19 pandemic could significantly influence the compliance with these healthy recommendations depending on the type of lockdown imposed on the country (e.g., total or partial lockdown).

The strength of this study is that it is the first study which compares COVID-19 restriction-related changes in 24-h movement behaviors, offering information from very high and high human development index countries. Some studies have pointed out differences between PA, ST and sleep duration (continuous variables). However, they did not report information about meeting all of the 24-h movement guidelines. This fact has a great relevance, since the need to identify geographic and country development differences influencing health behaviors in the youngest population in order to develop health promotion strategies has been pointed out [13,32]. Moreover, the present study includes a large sample of both Spanish and Brazilian preschoolers, children and adolescents with assessments of the variation in their health behaviors and meeting the 24-h movement guidelines. However, limitations could be reported. This study has not analyzed the geographical distribution of the sample nor the type of housing (e.g., with yard access, apartment with limited space) that could influence the level of physical activity.

## 5. Conclusions

The prevalence of preschoolers, children and adolescents in both the Spain and Brazil samples meeting the 24-h movement guidelines was low before the COVID-19 pandemic and even worse afterwards. This fact highlights the need to make efforts to protect and support healthy behaviors during a period of lockdown in young people. Identifying the proportion of preschoolers, children and adolescents meeting the 24-h movement guidelines during the COVID-19 pandemic lockdown provides evidence to inform strategies to avoid the potential harmful collateral effects precipitated by pandemic-related restrictions.

## Figures and Tables

**Figure 1 children-08-00083-f001:**
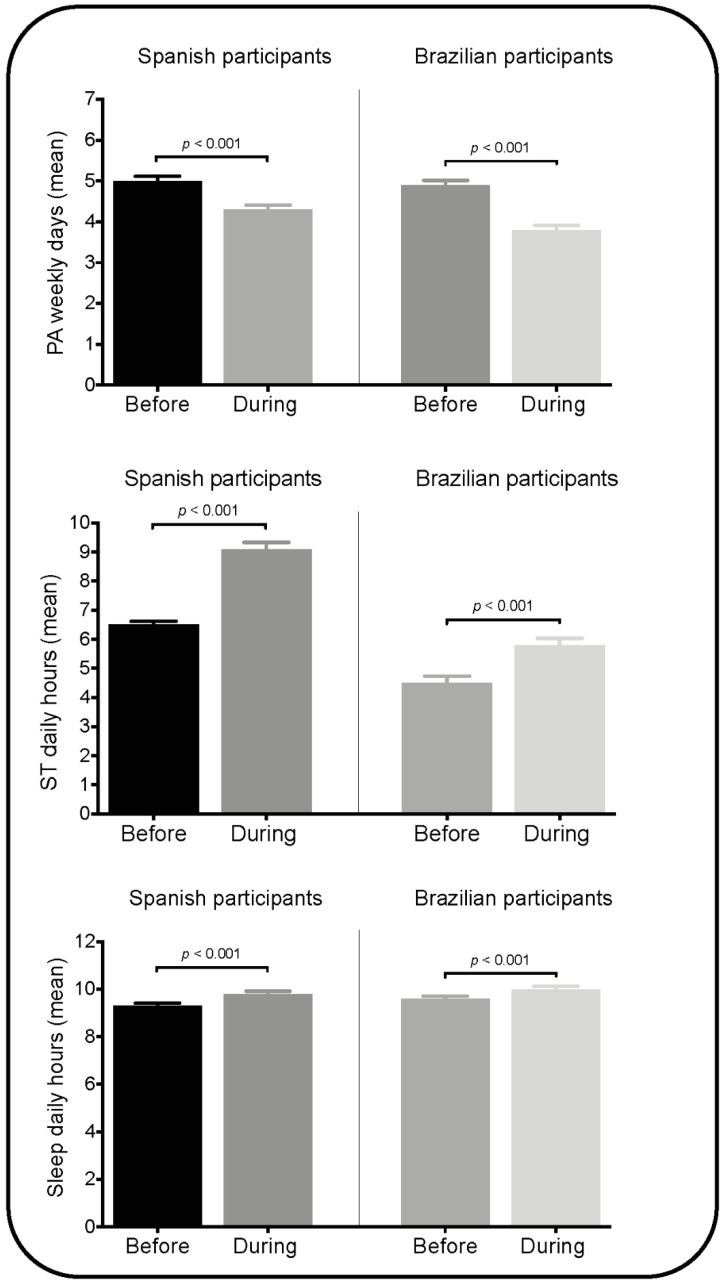
Changes in parental-reported physical activity (PA), screen time (ST) and sleep duration during the COVID-19 lockdown according to the country. Data expressed as mean (SEM).

**Figure 2 children-08-00083-f002:**
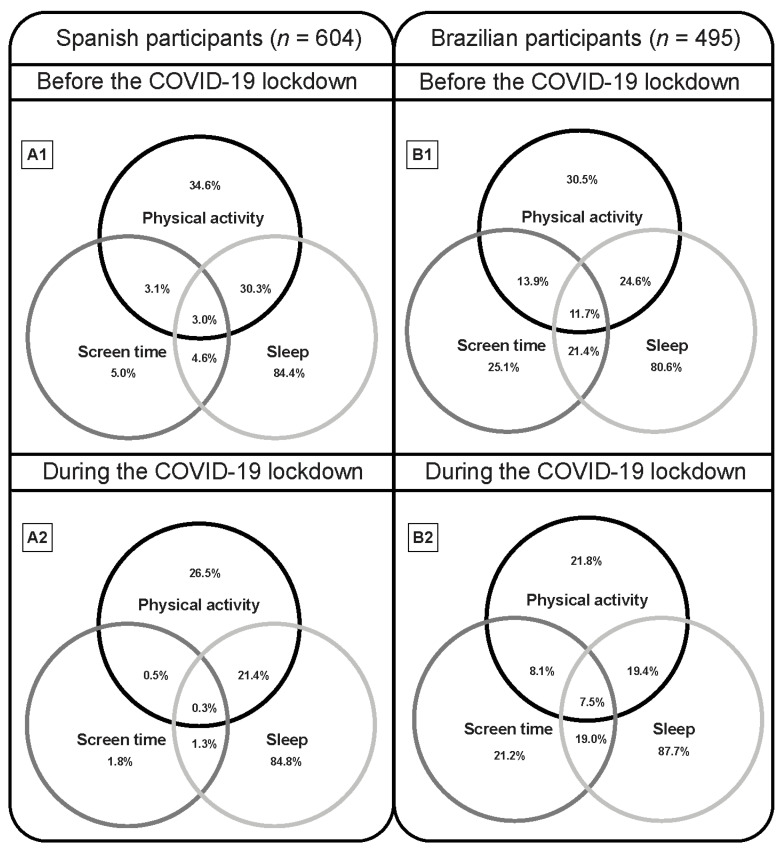
Venn diagram showing the proportion meeting the 24-h guidelines in both Spanish and Brazilian preschoolers, children and adolescents.

**Table 1 children-08-00083-t001:** Descriptive characteristics of the sample.

Variables	Total Sample (*n* = 1099)				
	Spain (*n* = 604)		Brazil (*n* = 495)		*p*
	M/n	SD/%	M/n	SD/%	
Age (years)	12.1	4.6	10.7	4.3	<0.001
Preschoolers	75	12.4	71	14.3	<0.001
Children	208	34.4	236	47.7	
Adolescents	321	53.1	188	38.0	
Sex					
Males	301	49.8	275	55.6	0.059
Females	303	50.2	270	44.4	
Breadwinner’s educational level					
Incomplete primary	22	3.6	28	5.7	<0.001
Complete primary education	91	15.1	6	1.2	
Incomplete secondary education	137	22.7	43	8.7	
Complete secondary education	133	22.0	27	5.5	
Incomplete higher education	34	5.6	40	8.1	
Complete higher education	187	31.0	351	70.9	
Socioeconomic status ^a^					
Low SES	161	26.7	201	40.6	<0.001
Medium SES	308	51.0	221	44.6	
High SES	135	22.4	73	14.7	
Anthropometric data					
Weight (kg)	43.8	20.1	46.2	20.4	0.053
Height (cm)	146.2	26.3	148.4	25.1	0.160
BMI (z-score)	0.9	1.8	0.9	1.8	0.560
Nutritional status ^b^					
Severe thinness	8	1.3	6	1.2	0.876
Thinness	11	1.8	10	2.0	
Normal weight	334	55.3	263	53.1	
Overweight	106	17.5	99	20.0	
Obesity	145	24.0	117	23.6	
Overweight/Obesity	251	41.5	216	43.6	0.540

Data expressed as a mean (M) and standard deviation (SD) for continuous variables or numbers and percentage for categorical variables. BMI: body mass index. SES: socioeconomic status. ^a^ Socioeconomic status according to the Family Affluence Scale (FAS-III) [19]. ^b^ Nutritional status according to the World Health Organization criteria [20,21].

## Data Availability

The data presented in this study are available on request from the corresponding author. The data are not publicly available because they belong to minors.

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
