# Peer review of "Changes in Healthy Behaviors and Meeting 24-h Movement Guidelines in Spanish and Brazilian Preschoolers, Children and Adolescents during the COVID-19 Lockdown"

_children, 2021, doi:10.3390/children8020083_

Round 1

Reviewer 1 Report

This is an interesting paper on the influence of COVID-19 measures on Health Behaviour of children. The application of the study in two different countries makes it even more interesting.

My main critical comment is that the comparison between 'before' and 'during' COVID are solely made on the basis of data collected during COVID. Were there no other sources of information about the compliance to the Guidelines?

Authors state (line 92) that the measure for PA was reliable and valid, with a reference to a paper from a COVID free period. I find it hard to believe that the answers to the questions:

  1. ‘‘Normally, how many days was your child physically active for a total of at least 60 min?’

  2. ‘‘During COVID, how many days was your child physically active for a total of at least 60 min?’

Are independently reliable and valid. I think that the perception of the effect of COVID is taken into account by the parents. So a comparison with independent data would be nice. If there is no data to compare to. Please discuss the issue depency/independency in relation to the validity.

(Being at home with your kids and seeing them behind a screen a lot can give you the idea that it must be more than normally etc. etc).

Suggestion:

You mention travel time (to school) in relation to sleep. Could it also play ar role in PA. Travelling might contribute to PA. So a diminishing PA might be explained by absence of travelling activities.

  1.  

Reviewer 2 Report

The authors have addressed an important heath behavior issue among children and adolescents amidst the Covid-19 pandemic crisis. Although it's hard to predict how impactful the behavior changes our children will take to shape their future behaviors post Covid-19, we can at least observe and analyze the current changes to provide strategies to enhance the healthy behaviors for better future development thanks to the research the authors conducted.  

The authors provided sufficient background and rationale for conducting the survey research. The results are well presented. The authors put a lot of thoughts in discussion with integration of studies from different countries. Some limitations were recognized, and the reasoning seems to be logical. 

A couple of questions/suggestions:

1) Was the geographic information obtained from the survey participants?  Were they city or rural dwellers from each country? Were they from big metropolitans or small towns? Could this be demonstrated in Material and Method or Results section since living environment will largely define the perimeter of physical activities?

2) In the discussion, the author did mention "... need to identify geographic and country development differences influencing health behaviors in the youngest population. (Line 241)". Housing environment, e.g., single family with yard access vs apartment with limited outdoor perimeter, and the distance to outdoor walking/jogging trails will significantly influence activity levels during the lockdowns. If authors have collected any geological information, further discussion would be enabled. If not, this may be addressed as the research limitation. 

3) Will the complete survey questionnaire be provided to readers and the rest of the research community?   
